# Mixed methods study to develop the content validity and the conceptual framework of the electronic patient-reported outcome measure for vascular conditions

Ahmed Aber  ,[1] Patrick Phillips,[1] Elizabeth Lumley,[1] Stephen Radley,[2] Steven M Thomas,[1,3] Shah Nawaz,[3] Georgina Jones,[4] Jonathan Michaels  ,[1] UK Vascular PROMs Group

[1]ScHARR, The University of Sheffield, Sheffield, UK
[2]Jessop Wing, Sheffield Teaching Hospitals NHS Foundation Trust, Sheffield, UK
[3]Sheffield Vascular Institute, Sheffield Teaching Hospitals NHS Foundation Trust, Sheffield, UK
[4]Leeds Social Sciences School, Leeds Beckett University, Leeds, UK

**Correspondence to**
Dr Ahmed Aber;
a.aber@sheffield.ac.uk

## ABSTRACT

**Objective** The aim of this paper is to describe the stages undertaken to generate the items and conceptual framework of a new electronic personal assessment questionnaire for vascular conditions.

**Design** A mixed methods study: First a survey of vascular clinicians was completed to identify the most common conditions treated in vascular clinics and wards. Quantitative systematic reviews were done to identify validated patient-reported outcome measures (PROMs) for direct inclsuion in the new instrument. However, due to scarcity of validated PROMs, the items of the new instrument were mainly based on a large qualitative study of patients and systematic reviews of the qualitative evidence . This was followed by a quantitative clinicians' consensus study and, finally, a qualitative face validity study with patients.

**Participants** Vascular patients participated in the primary qualitative study and the face validity study. In the qualitative study, 55 patients were interviewed, and for the face validity, 19 patients gave feedback. Twelve clinicians completed the survey and 13 completed two cycles of the clinicians' consensus study.

**Results** The items and scales in the electronic personal assessment questionnaire for vascular conditions (ePAQ-VAS) were generated based on the results of five systematic reviews evaluating existing PROMs for possible inclusion in ePAQ-VAS, five systematic reviews of qualitative evidence, a primary qualitative study involving 55 patients and clinicians' input. One hundred and sixty-eight items were initially generated, of which 59 were eliminated by the expert panel due to repetition. The instrument was divided into one generic and three disease-specific sections (abdominal aortic aneurysm, carotid artery disease and lower limb vascular conditions). In each section, items were grouped together into putative scales. Fifty-five items were grouped across eight scales; the remaining items were kept as individual items, because of relevance to service users.

**Conclusions** This multidimensional electronic questionnaire covers the most common vascular conditions. This is particularly important for patients

## Strengths and limitations of this study

► This electronic patient assessment questionnaire for vascular conditions was developed with input from patients and clinicians.
► The themes generated from previously published five comprehensive qualitative reviews and a qualitative study of vascular patients were used to develop the items.
► Vascular clinicians were surveyed to ensure clinically relevant conditions and questions were included.
► The burden of questionnaire is its main limitation; however, providing strict skipping rules, the patient were only be presented with the relevant sections and questions of the instrument.
► The face validity study examined the clarity and relevance of the items; however, the comprehensiveness of these PROMs was not assessed.

presenting with mixed symptoms or multiple conditions. This tool captures symptomatology, health related quality of life (HRQoL) and other clinically relevant data, such as experience with services and comorbidities.

## INTRODUCTION

Vascular conditions can cause problems throughout the body; epidemiological studies suggest that both venous and arterial diseases are very common.[1 2] It therefore makes sense to assess individuals with vascular disease holistically, investigating existing or potential manifestations of vascular disease and the impact of conditions on health-related quality of life. Patient-reported outcome measures (PROMs) are questionnaires or instruments, designed to elicit information directly from the patient and can be used as part of such an assessment.

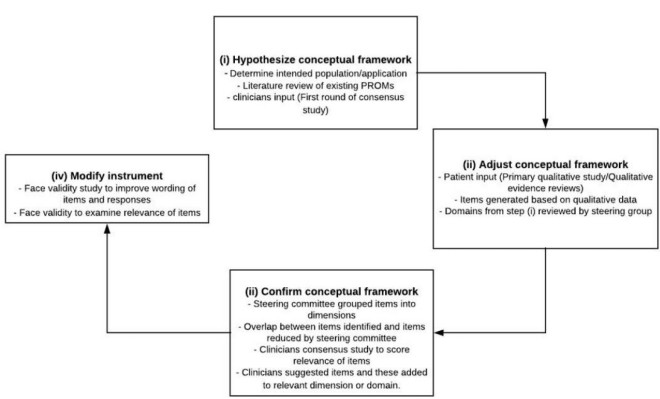

**Figure 1** Development of ePAQ-VAS conceptual framework. PROMs, patient-reported outcome measures; ePAQ-VAS, electronic personal assessment questionnaire for vascular patients.

Validity and reliability are integral to developing or selecting a PROMs. A key aspect of validity is *content validity*, and international guidelines including the US Food and Drug Administration (FDA) guidance stress the importance of this psychometric property.[3]

Many generic and condition specific PROMs have been adopted to examine impact of vascular conditions on patients and measure outcomes. This is despite a lack of evidence that they have been developed and evaluated in-line with accepted guidelines; in addition, these instruments are rarely used or formally evaluated in routine patient assessment in day-to-day clinical practice. We conducted scoping searches and informal discussions with vascular clinicians to identify any existing PROMs; however, these preliminary stages in the research process suggested an absence of valid and reliable PROMs for use in vascular populations.

In this paper, we report the stages in developing an electronic personal assessment questionnaire for vascular patients (ePAQ-VAS). This includes:

1. Identifying the main vascular conditions to be included in this electronic measure based on a survey of clinicians treating vascular disease.
2. Developing a hypothesised framework for the sections for different disease categories based on the previous systematic reviews that identified PROMs used in patients with abdominal aortic aneurysm (AAA), carotid artery disease (CAD), peripheral arterial disease (PAD), venous leg ulcers (VLU) and varicose veins (VV).[4–8]
3. Developing the items within each section of ePAQ-VAS based on qualitative systematic reviews[9–12] and a primary qualitative study.[13]
4. A consensus study with clinicians to rate the relevance of included items and to add items to ePAQ-VAS based on the opinion of vascular surgeons, radiologists and nurses.
5. A face validity study with vascular patients to examine the clarity and relevance of the items within ePAQ-VAS.

The aim of these steps was to develop a single electronic instrument covering most vascular conditions in line with international guidance.[3] The conceptual framework and items were developed in a way to ensure this assessment tool can be used in patients with mixed symptoms and multiple vascular conditions. Every patient to receive a unique voucher code along with their clinic letters. The code can be used to access and complete ePAQ-VAS at home or in the outpatient clinic using computers or other electronic devices.

The server of ePAQ is hosted and integrated with National Health Service (NHS) N3-based informatics systems. Other ePAQ questionnaires such as ePAQ-Pelvic floor and ePAQ-preassessment are in clinical use in different NHS hospitals. ePAQ Ltd is an NHS spin-out technology company, and the patient data collected by the company can be linked to the unique NHS number of each patient, and although there is a lack of integrated

**Table 1** Results from the systematic reviews of psychometric evaluation of vascular PROMs

| Condition | Number of citations | Number of included papers | Results | Conclusions |
|---|---|---|---|---|
| AAA | 1232 | 3 | 4 validated PROMs identified: 1 generic, 1 vascular generic and 2 condition specific | This review has highlighted a gap in the evidence for validated PROMs in AAA. Due to a lack of rigorous psychometric testing. |
| CAD | 1670 | 5 | 6 validated PROMs identified: 4 generic and 2 condition specific | There was a lack of validated PROMs to measure outcomes for CAD patients. |
| PAD | 6981 | 14 | 13 validated PROMs identified: 6 generic and 7 condition specific | VascQol was the most psychometrically robust instrument. |
| VV | 3879 | 7 | 3 validated PROMs identified: 1 generic and 2 condition specific | Aberdeen Varicose Vein Questionnaire is the most psychometrically robust disease-specific PROMs for use with VV patients. |
| VLU | 3879 | | 7 validated PROMs identified: 3 generic and 4 condition specific | The most valid and reliable condition specific PROMs was VLU-QOL. |

AAA, abdominal aortic aneurysm; CAD, carotid artery disease; PAD, peripheral arterial disease; VLU, venous leg ulcer; VLU-QOL, venous leg ulcer quality of life; VV, varicose veins.

**Table 2** Participant characteristics of the primary qualitative study

|  | AAA | CAD | PAD | VV | VLU | Total |
|---|---|---|---|---|---|---|
| Gender, n (%) | | | | | | |
| Male | 10 (77) | 5 (56) | 11 (79) | 5 (50) | 8 (80) | 39 (70) |
| Female | 3 | 4 | 3 | 5 | 2 | 17 |
| Age range (mean) | 53–87 (72) | 52–86 missing | 47–82 (69) | 35–77 (50) | 47–84 (59) | 35–87 missing |

AAA, abdominal aortic aneurysm; CAD, carotid artery disease; PAD, peripheral arterial disease; VLU, venous leg ulcers; VV, varicose veins.

digital infrastructure in the NHS, the technology is available for future use to link records collected by different NHS providers.

## METHODS

Clinicians involved in the care of vascular patients were invited to identify the common vascular conditions treated by vascular surgeons and vascular specialists. They were asked to list the key issues, symptoms and the impact of these conditions on patients suffering with these diseases. Data from this round were used to inform qualitative evidence synthesis.

The conceptual framework of ePAQ-VAS was based on primary qualitative interviews with vascular patients, input from clinicians, systematic reviews examining the validity of existing PROMs and qualitative reviews of the impact of vascular diseases on quality of life. Figure 1 illustrates the process used to develop ePAQ-VAS in accordance to s guidelines.[3]

### Systematic reviews to identify and appraise existing PROMs

Systematic searches were conducted of bibliographic databases including CINAHL via EBSCO, MEDLINE and MEDLINE in Process via Ovid, Embase via Ovid, PsycINFO via Ovid, Social Science Citation Index/Science Citation Index via Web of Science (Thomson Reuters) and Proquest dissertations and theses. PROMs were included where there was evidence that they had undergone some form of psychometric evaluation that would allow the validity, reliability and responsiveness of the PROMs to be assessed. Included PROMs were categorised per type (generic or condition specific) and the vascular population(s) in which they had been validated. Quality assessment[3][14] was conducted to identify high-quality existing PROMs for possible direct inclusion in ePAQ-VAS or to be used as a basis to inform the qualitative evidence synthesis. For further information about the appraisal criteria to examine the robustness of the psychometric analysis and samples of search strategies, please see the online supplementary materials.

**Table 3** Findings from the primary qualitative study with vascular patients

| Condition | Sample size | Key findings |
|---|---|---|
| AAA | 13 | No physical symptoms, a small number of participants reported abdominal pain and pain in their legs. Uncertainty, anxiety and fear of rupture and death appeared to impact most greatly on people's QoL. |
| CAD | 9 | This condition seemed to have had the least impact on physical and social function, although psychologically it created a sense of worry and anxiety for some participants. The main reported outcome was fear of having a major stroke. |
| PAD | 14 | Pain and mobility were the most commonly reported themes. The extent to which they impacted on QoL was associated with the severity, age expectations and social support. Fear of the symptoms worsening and amputation was evident. |
| VV | 10 | VV do not appear to have had a major impact on overall QoL for most the participants. Pain was the most common issue. The perceived unpleasant appearance of the VV seemed to have the greatest psychological impact. Many of the participants had had their VV for very long periods of time, often just 'putting up with it' for numerous years before seeking help. |
| VLU | 10 | The impact of VLU on QoL differed within the group. For some, there were no major issues, and having a VLU was accepted as part of their current life, with the hope that it would heal eventually. For others, there was a far more significant effect. Pain was quite severe for some participants leading to a significantly reduced QoL. VLU appeared to have a significant psychological impact causing a high degree of distress for some. |

AAA, abdominal aortic aneurysm; CAD, carotid artery disease; PAD, peripheral arterial disease; QoL, quality of life; VLU, venous leg ulcer; VV, varicose veins.

**Table 4** Map of symptoms and quality of life concepts across five conditions

| | PAD | AAA | CAD | VV | VLU |
|---|---|---|---|---|---|
| **Symptoms** | | | | | |
| No symptoms | | × | × | | |
| Pain | × | × | × | × | × |
| Neck pain | | | × | | |
| Leg pain | × | × | × | × | × |
| Abdominal pain | × | × | | | |
| Arm pain | × | | | | |
| Cramp/aching | × | × | | × | × |
| Burning sensation | | | | | × |
| Pain severity | × | × | × | × | × |
| Pain on walking | × | × | | × | × |
| Pain at rest | × | | | × | |
| Pain when standing | | | | × | × |
| Mobility | × | × | × | × | × |
| Distance | × | × | | × | × |
| Speed | × | × | | | |
| Stairs/slopes | × | × | | | |
| Non-healing wounds | × | | | | × |
| Comorbidities | × | × | × | | × |
| Progression of symptoms | × | × | | × | × |
| Sleep | × | | × | × | × |
| Swelling | | | × | × | |
| Loss of balance | | | × | | |
| Confusion | | | × | | |
| **Impact on physical functioning** | | | | | |
| Hobbies | × | | | × | × |
| Exercise | × | × | | × | × |
| Daily activities | × | | | | × |
| **Social impact** | | | | | |
| Travel | × | × | × | | |
| Social activities | × | × | | × | × |
| Social support | × | × | × | | |
| **Psychological impact** | | | | | |
| Anxiety | × | × | × | × | × |
| Depression | × | × | | | × |
| Feelings of loss | × | | | × | × |
| Health expectations | × | × | × | × | × |
| Unsightly appearance | | | | × | |
| Feeling self-conscious | × | | | × | × |
| Fear of worsening symptoms | × | × | × | × | × |
| Fear of rupture death | | × | | | |
| Fear of amputation | | | | × | × |
| Fear of stroke | | | × | | |
| **Financial impact** | | | | | |
| Income | × | × | | | × |
| Time off work | | | | × | × |
| **Lifestyle** | | | | | |
| Smoking | × | × | × | × | × |
| Exercise | × | × | × | × | |
| Diet | × | × | × | | × |
| Weight | | | | × | × |

AAA, abdominal aortic aneurysm; CAD, carotid artery disease; PAD, peripheral arterial disease; VLU, venous leg ulcer; VV, varicose veins.

## Primary qualitative study

Semistructured interviews were conducted with 55 vascular patients from Sheffield Teaching Hospitals NHS Foundation Trust following purposeful sampling to ensure a range of participants of different age and gender, at different stages of treatment and covering the main five vascular conditions (AAA, PAD, CAD, VLU and VV). A consultant vascular surgeon or specialist nurse approached each patient either in clinic or over the telephone to explain about the project and ask if the patient would be interested in participating in the study. If the initial approach was by the clinician in clinic, the researcher would then speak to the patient and give further information about the project including a participant information sheet (PIS) before taking contact details. For those patients who were first contacted over the phone, the clinician would then gain verbal consent to pass on their contact details to a researcher. Copies of the PIS were sent out through the mail to those who had not been initially approached in clinic. The researcher gave at least 24 hours for the patient to read through the PIS and consider the information before contacting each person by telephone to ask if they would be interested in participating in an interview. If they were interested in taking part, a date and time was agreed for a researcher to visit the participant at home to carry out an interview. Questions were asked about the signs, symptoms and impact of the condition on function and lifestyle. On the day of the interview, the trained qualitative researcher checked if the participant understood the PIS and took informed written consent. Field notes

**Table 5** Results from qualitative reviews examining the impact of the major vascular conditions on quality of life

| Condition | Numbers of citations | Number of included studies | Key themes |
|---|---|---|---|
| AAA | 315 | 3 | Anxiety and *lack* of physical symptoms. |
| CAD | 964 | 3 | Symptoms, psychological and social impact, risk and service experience. |
| PAD | 973 | 9 | Pain, compromised physical function and impact on social life. |
| VV | 1804 | 3 | Adaptation – coping strategies employed to limit various impacts, appearance of VV. |
| VLU | 1804 | 13 | Pain, odour and exudate – impact on sleep, mobility and mood. |

AAA, abdominal aortic aneurysm; CAD, carotid artery disease; PAD, peripheral arterial disease; VLU, venous leg ulcer; VV, varicose veins.

were taken to aid interpretation of the interview data. Each interview was recorded and transcribed verbatim. Personal details were removed from the transcript to enhance participant anonymity. The interview transcripts were typed and uploaded into NVIVO V.11 (QSR International, Warrington, UK) for management and analysis.

### Systematic reviews of the qualitative evidence

Systematic searches of the following databases; CINAHL via EBSCO, MEDLINE and MEDLINE in Process via Ovid, Embase via Ovid, PsycINFO via Ovid, Social Science Citation Index/Science Citation Index via Web of Science (Thomson Reuters) and Proquest dissertations were conducted to identify existing qualitative research detailing vascular patients' experience of living with AAA, PAD, CAD, VLU and VV. For further samples of search strategies, please see the online supplementary materials.

### Analysis of the qualitative evidence

Qualitative data from the primary study and each of the systematic qualitative reviews were analysed separately. Framework analysis was used to analyse the interviews.[15] This analysis includes five stages:

► The first stage involved familiarisation by reading of the transcripts and reading the primary data.
► The second stage involved identification of a thematic framework; the thematic framework was based either on clinical opinion for areas with no valid PROMs, such as AAA and CAD, or a combination of clinical opinion and, when available, the scales of PROMs with good content validity.
► In the third stage, the data were coded and indexed by applying the thematic framework to the whole data set until saturation was achieved. An second researcher checked all the themes that were identified, and differences in were discussed and adjusted involving a third senior author (GJ).
► At the fourth stage, a framework matrix was created by arranging the data per the thematic references.
► Finally, mapping and interpretation, including examining patterns within the data and associations with it.

### Clinicians' input and consensus exercise

Twenty-three clinicians involved in the care and management of patients with vascular conditions were invited to a survey to list the most common vascular conditions managed by them and to list the key issues, symptoms and the impact of these diseases on patients. Data from this round were used to inform qualitative evidence synthesis.

Different group of clinicians involved in the care of vascular patients were invited to a consensus study to score the relevance of items (questions) in the provisional version of ePAQ-VAS. In total, 30 clinicians including vascular surgeons, interventional radiologists, vascular nurses, physiotherapists and occupational therapists were invited. Participants were asked to rate the appropriateness of each question on a 5-point Likert scale of 'strongly disagree' (=0) to 'strongly agree' (=4). This process was repeated, and members of the clinicians' panel were presented with the aggregate findings of the previous round and again asked to score each question. This process aimed to examine the relevance of each item from the clinicians' perspective and to identify any new items suggested by the clinicians.[16]

### Developing scales and items

The ePAQ development team (AA, EL, PP, GJ and SR) employed an iterative process, incorporating evidence from the systematic reviews, qualitative study and the clinicians' consensus study. In line with the FDA guidance,[3] items (questions) were developed from the qualitative data using the following three steps: interpretation, translation and triangulation of themes.

Interpretation involved familiarisation with the language used in the primary data included in the synthesis. This enabled translation of descriptions of apparently diverse issues affecting vascular patients into a single set of harmonised themes. The resulting themes were used to develop the items for ePAQ-VAS. The items were grouped into sections, and each section further divided into scales consisting of a connected group of items. Triangulation was performed across evidence sources to ensure the items comprehensively covered all issues of importance to patients with AAA, PAD, CAD, VLU and VV.

**Table 6** Structure of the main ePAQ-VAS with evidence base for inclusion of individual items, scales and sections

| Section | Scale | Clinicians' consensus study | Most valid condition specific PROMs | | | Qualitative study | | | | | Qualitative reviews | | | | | Question text |
|---|---|---|---|---|---|---|---|---|---|---|---|---|---|---|---|---|
| | | | VLU-QOL | PADQOL | AVVQ | AAA | CAD | VV | VLU | PAD | AAA | CAD | VV | VLU | PAD | |
| Generic | Pain | 4 | × | × | × | × | × | × | × | × | × | × | × | × | × | Do you suffer with any pain? |
| Generic | Pain | 4 | × | – | – | × | × | × | × | × | × | × | × | × | × | Use the image below and click on body parts where you experience pain or discomfort. |
| Generic | Pain | – | × | – | – | × | × | × | × | × | × | × | × | × | × | Please use your own words to describe this problem. |
| Generic | Pain | 4 | × | × | × | × | × | × | × | × | × | × | × | × | × | How often do you experience a significant amount of pain? |
| Generic | Pain | 4 | × | – | – | × | × | × | × | × | × | × | × | × | × | How much do problems caused by pain affect your overall enjoyment of life? |
| Generic | Sensation | 4 | – | – | – | | × | × | | × | × | × | × | | × | Do you experience any numbness or pins and needles in any part of your body? |
| Generic | Sensation | 4 | – | – | – | | × | × | | × | × | × | × | | × | Please use the image below to select where you experience sensation change in your body. |
| Generic | Sensation | – | – | – | – | | × | × | | × | × | × | × | | × | Please use your own words to describe this problem. |
| Generic | Sensation | 4 | – | – | – | | × | × | | × | × | × | × | | × | How often do you experience numbness or pins and needles? |
| Generic | Sensation | 4 | – | – | – | | × | × | | × | × | × | × | | × | How much do problems caused by numbness or pins and needles affect your overall enjoyment of life? |
| Generic | Weakness | 4 | – | – | – | | | | | | × | × | | | × | Do you have any loss of strength or weakness in any part of your body? |
| Generic | Weakness | 3 | – | – | – | | | | | | × | × | | | × | Please use the image below to indicate the areas where you experience any physical weakness. |
| Generic | Weakness | – | – | – | – | | | | | | × | × | | | × | Please use your own words to describe this problem. |
| Generic | Weakness | 4 | – | – | – | | | | | | × | × | | | × | How often do you experience loss of strength or weakness? |
| Generic | Weakness | 3 | – | – | – | | | | | | × | × | | | × | How much do problems caused by weakness affect your overall enjoyment of life? |

Continued

**Table 6** Continued

| Section | Scale | Clinicians' consensus study | Most valid condition specific PROMs | | | Qualitative study | | | | | Qualitative reviews | | | | | Question text |
|---|---|---|---|---|---|---|---|---|---|---|---|---|---|---|---|---|
| | | | VLU-QOL | PADQOL | AVVQ | AAA | CAD | VV | VLU | PAD | AAA | CAD | VV | VLU | PAD | |
| Carotid | Anxiety | 4 | | | | | × | | | | | × | | | | Do you worry about having a stroke? |
| Carotid | Anxiety | 3 | | | | | × | | | | | × | | | | Does carotid artery disease make you feel anxious? |
| Carotid | Anxiety | New item | | | | | | | | | | | | | | Are you worried about your health getting worse because of carotid artery disease? |
| Carotid | Anxiety | New item | | | | | | | | | | | | | | Are you worried about losing your independence because of carotid artery disease? |
| Carotid | Symptoms | 2 | | | | | × | | | | | | | | | Do you have any problems with maintaining your balance? |
| Carotid | Symptoms | 2 | | | | | × | | | | | | | | | Do you suffer with any problems with your memory? (eg, forgetting or losing things) |
| Carotid | Symptoms | 4 | | | | | × | | | | | × | | | | Have you had any problems with your speech? (eg, slurring your words or not being able to speak or say things properly) |
| Carotid | Symptoms | New item | | | | | | | | | | | | | | Do you have any problems with swallowing food? |
| Carotid | Symptoms | 4 | | | | | × | | | | | × | | | | Have you had any problems with partial or complete loss of vision in either of your eyes? |
| Carotid | Symptoms | 4 | | | | | × | | | | | × | | | | How would you describe any loss of vision in your right eye? |
| Carotid | Symptoms | 4 | | | | | × | | | | | × | | | | How would you describe any loss of vision in your left eye? |
| Carotid | ADL | 3 | | | | | × | | | | | × | | | | How much do problems caused by carotid artery disease (anxiety associated with diagnosis or physical symptoms) affect your overall enjoyment of life? |

**Table 6** Continued

| Section | Scale | Clinicians' consensus study | Most valid condition specific PROMs | | | Qualitative study | | | | | Qualitative reviews | | | | | Question text |
|---|---|---|---|---|---|---|---|---|---|---|---|---|---|---|---|---|
| | | | VLU-QOL | PADQOL | AVVQ | AAA | CAD | VV | VLU | PAD | AAA | CAD | VV | VLU | PAD | |
| Carotid | ADL | 4 | | | | | × | | | | | × | | | | How much do problems caused by carotid artery disease (eg, mini-stroke or stroke memory balance speech visual or other related issues) affect your physical activities such as exercise walking or running? |
| Carotid | ADL | 3 | | | | | × | | | | | × | | | | How much do problems caused by carotid artery disease (eg, mini-stroke or stroke memory balance speech visual or other related issues) affect your ability to undertake personal roles and responsibilities such as caring for others study or work? |
| Carotid | ADL | 4 | | | | | × | | | | | × | | | | How much do problems caused by carotid artery disease (eg, mini-stroke or stroke memory balance speech visual or other related issues) affect your ability to look after yourself? |
| Carotid | ADL | 3 | | | | | × | | | | | × | | | | How much do problems caused by carotid artery disease (eg, mini-stroke or stroke memory balance speech visual or other related issues) affect your social activities such as visiting friends and family? |
| Carotid | ADL | 2 | | | | | × | | | | | × | | | | How much do problems caused by carotid artery disease visual or other related issues) affect your mood? |
| AAA | Symptoms | 4 | | | | × | | | | | × | | | | | Do you have any abdominal (tummy) pain? |
| AAA | Symptoms | New item (3) | | | | | | | | | | | | | | Do you experience a throbbing feeling in your abdomen (tummy)? |
| AAA | Anxiety | 4 | | | | × | | | | | × | | | | | Do you worry about aortic aneurysm? |
| AAA | Anxiety | 4 | | | | × | | | | | × | | | | | Do you worry about any symptoms you experience that may be caused by aortic aneurysm? |

Continued

**Table 6** Continued

| Section | Scale | Clinicians' consensus study | Most valid condition specific PROMs | | | Qualitative study | | | | | Qualitative reviews | | | | | Question text |
|---|---|---|---|---|---|---|---|---|---|---|---|---|---|---|---|---|
| | | | VLU-QOL | PADQOL | AVVQ | AAA | CAD | VV | VLU | PAD | AAA | CAD | VV | VLU | PAD | |
| AAA | Anxiety | 4 | | | | × | | | | | × | | | | | Do you worry about possible increase in the size of your aneurysm? |
| AAA | Anxiety | 4 | | | | × | | | | | × | | | | | Do you fear sudden death or rupture of your aortic aneurysm? |
| AAA | Anxiety | 4 | | | | × | | | | | × | | | | | Do you avoid physical exertion because of having an aortic aneurysm? |
| AAA | Anxiety | 3 | | | | × | | | | | × | | | | | Do you avoid travelling independently because of aortic aneurysm? |
| AAA | ADL | 3 | | | | | | | | | × | | | | | How much do problems caused by aortic aneurysm affect your overall enjoyment of life? |
| AAA | ADL | 4 | | | | × | | | | | × | | | | | How much does aortic aneurysm affect your physical activities? (eg, exercise walking or going out) |
| AAA | ADL | 3 | | | | × | | | | | × | | | | | How much does aortic aneurysm affect your ability to undertake personal roles and responsibilities? (eg, caring for others study or work? |
| AAA | ADL | 3 | | | | | | | | | × | | | | | How much do you feel aortic aneurysm affects your ability to look after yourself? (eg, rest wash toilet or feed yourself) |
| AAA | ADL | 4 | | | | | | | | | × | | | | | How much does aortic aneurysm affect your social activities? (eg, visiting friends or family) |
| AAA | ADL | 4 | | | | × | | | | | × | | | | | Do you suffer from low mood because of having an aortic aneurysm? |
| LL | Ischaemic Pain | 4 | ± | ± | | | | × | × | × | | | × | | × | Do you experience any cramping pain in your legs or feet? |
| LL | Ischaemic Pain | 4 | ± | ± | | | | × | | × | | | | | × | Do you experience cramping pain in your legs or feet when walking? |
| LL | Ischaemic Pain | 4 | | ± | | | | | | × | | | | | × | How far can you walk before you experience any cramping pain in your legs or feet? |

Continued

**Table 6** Continued

| Section | Scale | Clinicians' consensus study | Most valid condition specific PROMs | | | Qualitative study | | | | | Qualitative reviews | | | | | Question text |
|---|---|---|---|---|---|---|---|---|---|---|---|---|---|---|---|---|
| | | | VLU-QOL | PADQOL | AVVQ | AAA | CAD | VV | VLU | PAD | AAA | CAD | VV | VLU | PAD | |
| LL | Ischaemic Pain | 4 | ± | | | | | | | × | | | | | × | Do you walk more slowly than you would choose to in order to avoid cramping pain in your legs and feet? |
| LL | Ischaemic Pain | 4 | − | | | | | | | × | | | | | × | Do you experience cramping pain in your legs or feet when walking uphill? |
| LL | Ischaemic Pain | 4 | − | | | | | | | × | | | | | × | Do you experience pain in your legs or feet when you climb stairs? |
| LL | Ischaemic Pain | 4 | − | | | | | × | × | × | | | × | × | × | Do you experience pain in your feet at night? |
| LL | Ischaemic Pain | 4 | − | | | | | | | | | | | | | Do you dangle one or both of your legs over the side of the bed to help reduce foot pain? |
| LL | Ischaemic Pain | 4 | − | | | | | × | × | × | | | × | × | × | Do you experience severe pain in your legs or feet when you are resting or sitting? |
| LL | Ischaemic Pain | 2 | − | | | | | | | × | | | | | × | Are you troubled by cold feet? |
| LL | Ulcer | 4 | X | − | | | | × | × | × | | | × | × | × | Have you ever had any ulcers on your legs or feet now or at any time in the past? |
| LL | Ulcer | 4 | − | − | | | | × | × | × | | | × | × | × | Please use the image below to show where you currently have any leg or foot ulcers. |
| LL | Ulcer | 4 | − | − | | | | | × | | | | | × | | Are you concerned about the smell of your leg ulcers? |
| LL | Ulcer | 4 | × | − | | | | | | | | | | × | | Are you concerned about the appearance of your leg ulcers? |
| LL | Ulcer | 4 | − | − | | | | | | | | | | × | | Do you have leg ulcers that leak fluid (watery liquid)? |
| LL | Ulcer | 4 | − | − | | | | | | | | | | × | | Do you experience infections in your leg ulcers? (eg, foul smell or pus) |
| LL | Ulcer | 4 | − | − | | | | | × | | | | | × | | Do you experience repeated leg ulcers? |

**Table 6** Continued

| Section | Scale | Clinicians' consensus study | Most valid condition specific PROMs | | | Qualitative study | | | | | Qualitative reviews | | | | | Question text |
|---|---|---|---|---|---|---|---|---|---|---|---|---|---|---|---|---|
| | | | VLU-QOL | PADQOL | AVVQ | AAA | CAD | VV | VLU | PAD | AAA | CAD | VV | VLU | PAD | |
| LL | Ulcer | 4 | × | – | × | | | | × | | | | | × | × | Do you worry about your leg ulcers? (eg, not healing becoming infected losing part of your leg or foot). |
| LL | VV | 4 | | – | – | | | × | | | | | × | | | Do you experience any bleeding from veins in your legs or feet? |
| LL | VV | 4 | | – | × | | | × | | | | | × | | | Do you have any problems with the skin over your varicose veins? |
| LL | VV | 4 | | – | × | | | × | | | | | × | | | Do varicose veins make you feel self-conscious or embarrassed? |
| LL | VV | 4 | × | – | × | | | × | | | | | × | × | × | Do leg or foot problems affect what clothing or shoes you can wear? |
| LL | VV | 4 | | – | × | | | × | | | | | × | × | | Do you experience any swelling in your legs or feet? |
| LL | VV | 4 | | – | × | | | × | | | | | × | × | × | Do you experience itching in your legs or feet? |
| LL | VV | 4 | | – | × | | | × | | | | | × | × | | Do you wear compression stockings or tights for your legs? |
| LL | Tissue loss | 4 | | – | – | | | | × | × | | | | | × | Have you lost any part of your legs or feet through amputation or gangrene? |
| LL | Tissue loss | 4 | | | – | | | | × | × | | | | | × | Please click on the appropriate part or parts of your legs feet or toes that you have had amputated or have been lost. |
| LL | Anxiety | 3 | × | × | × | | | × | × | × | | | × | × | × | Do you worry about your leg problems getting worse in the future? |
| LL | ADL | 3 | ± | × | ± | | | × | × | × | | | × | × | × | How much do leg or foot problems affect your overall enjoyment of life? |
| LL | ADL | 3 | ± | × | – | | | × | × | × | | | × | × | × | How much do leg or foot problems affect your ability to carry out physical activities? (eg, walking housework or exercise) |
| LL | ADL | 3 | ± | × | ± | | | × | × | × | | | × | × | × | How much do leg or foot problems affect your personal responsibilities? For example, caring for others study or work. |

Continued

**Table 6** Continued

| Section | Scale | Clinicians' consensus study | Most valid condition specific PROMs | | | Qualitative study | | | | | Qualitative reviews | | | | | Question text |
|---|---|---|---|---|---|---|---|---|---|---|---|---|---|---|---|---|
| | | | VLU-QOL | PADQOL | AVVQ | AAA | CAD | VV | VLU | PAD | AAA | CAD | VV | VLU | PAD | |
| LL | ADL | 3 | – | × | ± | | | × | × | × | | | × | × | × | How much do leg or foot problems affect your ability to look after yourself? (eg, rest wash toilet or feed yourself) |
| LL | ADL | 3 | ± | × | ± | | | × | × | × | | | × | × | × | How much do leg or foot problems affect your social activities? (eg, going out visiting friends or family) |
| LL | ADL | 4 | Xx | × | – | | | × | × | × | | | × | × | × | Do you suffer from low mood because of leg or foot problems? |
| Generic | Personal data | | – | × | – | × | × | | × | × | × | × | | × | × | Do you rely on any people to help you with your everyday activities? |
| Generic | Personal data | | – | | | × | | × | × | × | × | | × | × | × | Do you experience financial problems because of your vascular condition? |

The items of the eight scales are coloured in this table.

AAA, abdominal aortic aneurysm; ADL, activity of daily living; AVVQ, Aberdeen Varicose Veins Questionnaire; CAD, carotid artery disease; LL, lower limb; PAD, peripheral arterial disease; PADQOL, peripheral arterial disease quality of life; VLU, venous leg ulcer; VLU-QOL, venous leg ulcer quality of life; VV, varicose veins.

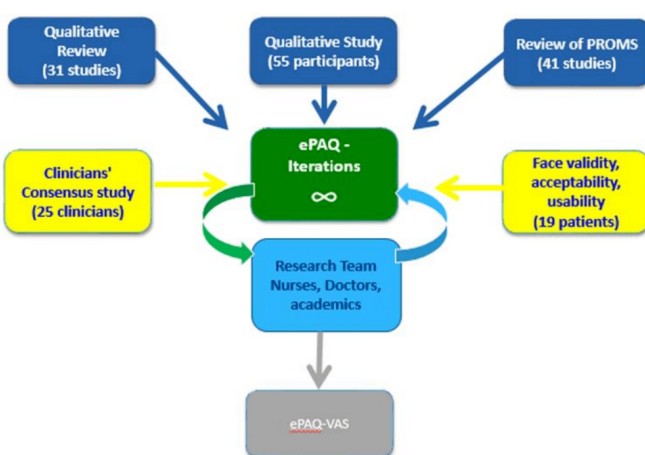

**Figure 2** Evidence synthesis to develop of ePAQ-VAS. ePAQ-VAS, electronic personal assessment questionnaire for vascular patients; PROMs, patient-reported outcome measures.

## Face validity of ePAQ-VAS

A second phase of semistructured patient interviews was conducted by (EL and PP) with 19 participants, purposefully sampled from the vascular populations previously described. This sample included patients with AAA, CAD, PAD, VLU and VVs. ePAQ-VAS (version 1) was presented to these patients, and a focused interview was conducted to investigate vascular patients' perceptions of the questionnaire in its entirety as well as the relevant items to the individual being interviewed. Questions were asked under the following headings of:

▶ Overall impressions.

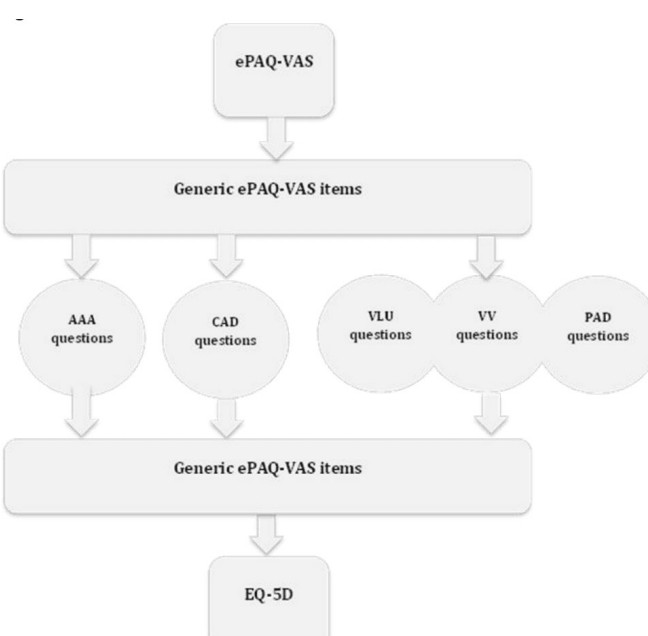

**Figure 3** Overview of ePAQ-VAS structure. AAA, abdominal aortic aneurysm; CAD, carotid artery disease; ePAQ-VAS, electronic personal assessment questionnaire for vascular patients; PAD, peripheral arterial disease; VLU, venous leg ulcers; VV, varicose veins.

▶ Clarity.
▶ Relevance and emotional response.

Interviews were audio taped, transcribed and analysed. A pragmatic approach was used for the analysis, with comments collated and presented back to the working group who made consensus decisions on revisions to ePAQ-VAS. Written consent was obtained from the participants.

### Patient and public involvement

The research question and output were developed in consultation with patients and public. The authors would like to thank the Cardiovascular Research Patient Panel at Sheffield Teaching Hospitals NHS Foundation Trust. The aim of the research was to develop a patient focused outcome measure. In this process, patients were recruited for two qualitative studies to ensure content validity and face validity of this tool. Patients were involved in every stage of the development of the study. The developed ePAQ-VAS has been used by patients in a clinical study, and there are plans for regular clinical use. The results will also be disseminated in relevant meetings and among patient groups.

## RESULTS

In total, 12 clinicians completed the first survey and identified PAD, AAA, VLU, VV and CAD as the most common vascular conditions treated by them. They listed common issues such as pain on walking, rest pain, reduced mobility or lack of mobility for patients with PAD and no physical symptoms for those with AAA but need for multidisciplinary approach to manage these patients. For patients with VLU, the main issues included burning pain, recurrence and healing; for patients suffering with VV, skin changes, appearance of leg and ulcer as well as ache were the main issues raised. The clinicians felt the key issue for patients with CAD was identifying patients benefiting from intervention and reducing the risk of stroke. The result from this survey was used to inform the analysis of qualitative data used to develop ePAQ-VAS.

Systematic reviews and assessment of psychometric evaluation were conducted for PROMs validated for use in PAD, AAA, VLU, VV and CAD. A total of 33 PROMs that had undergone some form of validation were identified in 41 studies (table 1).

No PROMs were identified that had undergone sufficiently rigorous development and validation to suggest that they were suitable for direct use in ePAQ-VAS, the details of these reviews have been reported previously[4–8]. Where evidence existed, this fell short of required standards.[3 14] For instance, the review investigating VV PROMs[4] found some evidence for, and discussion of, content validity in relation to the Aberdeen Varicose Vein Questionnaire (AVVQ) and suggest that it is the most appropriate existing condition-specific measure for use in a VV population. However, item generation for these PROMs involved a literature review and assessment

by clinicians of relevance of included items with no direct involvement of patients, therefore suggesting a deficiency in terms of content validity.[17] The scales from these reviews were used to provide a framework for the systematic qualitative reviews and the primary qualitative study.[5 9–13]

In total, 111 patients were approached, but only 55 patients (69.1% male) were interviewed about their experience of living with vascular disease, ages ranged from 35 to 77 years. For further information about the study participants, please see table 2.

Six overarching themes relating to the impact of the five vascular conditions were identified. These were symptoms (including pain), impact on physical function, social impact, psychological impact, financial impact and lifestyle. Pain and mobility were the most commonly reported themes by participants with PAD. The extent to which they impacted daily living was dependent on the severity of the disease, age expectations and social support. Fear of symptoms worsening and future amputation had a significant impact on daily living.

Most participants with AAA reported having no physical symptoms; a small number of participants reported abdominal pain and pain in their legs. Uncertainty, anxiety and fear of sudden death had the most impact on their quality of life. This was similar for patients with CAD who reported few lasting symptoms since the majority had what they described as a 'mini-stroke'. However, CAD patients reported the widest range of signs and symptoms, with nine different symptoms. This condition had the least impact on physical and social function, although psychologically it caused a sense of worry and anxiety. This was mainly caused by fear of having a major stroke.

Pain was the most common issue reported by patients with VVs; other issues included swelling of the legs and the impact of this on mobility. The perceived unpleasant appearance of the VV seemed to have had the greatest psychological impact and was described by many of the participants. The impact of VLU on daily living and quality of life differed within the group that was interviewed. For some, there were no major issues, and having a VLU was accepted as part of their life, with the hope that it would heal eventually. For others, there was a far more significant effect with reports of severe sharp pain that significantly reduced their quality of life. This had a bearing on people's mobility and their ability, or desire, to go out and socialise. Sleep was also disturbed due to pain. The progression of VLU had resulted in participants suffering for long periods of time. In addition, the non-healing or reoccurring nature of the condition had a significant impact for many. VLU appeared to have a significant psychological impact causing a high degree of distress for some patients. Summary results are shown in table 3.

Identified signs, symptoms and impact of the conditions were then mapped and tabulated to see which themes were relevant to which condition and where the similarities and differences lay (table 4).

A total of 31 studies were included across the five reviews of existing qualitative research.[6 10–13] A short summary of the main themes to emerge for each condition is shown in table 5.

The themes from the first round of the clinicians' consensus study, as well as scales of identified PROMs, were used to inform the framework analysis of the qualitative data. Items from existing PROMs were then mapped against emerging themes from the qualitative study, and the qualitative review synthesis for each condition, to explore which PROMs items or scales captured themes deemed to be the most pertinent to patients. A triangulation approach was followed, whereby researchers evaluated whether the concepts were the same (agreement), offered similar concepts (partial agreement), were in contradiction (dissonance) or were not present (silence). An example of this triangulation approach is provided in the online supplementary material. The results of the triangulation study were only used to group symptoms together and avoid repetition. No items were deleted based on the triangulation.

The ePAQ-VAS development team used the findings from the triangulation for AAA, PAD, VV, VLU and CAD to develop themes for distinct sections relevant for each of these vascular conditions. The primary qualitative data were used to create each item. Items were then grouped into sections, and within each section, there were scales consisting of items that measured the same latent variable such as anxiety related to the diagnosis of AAA. The results of the clinicians' consensus study were considered to add further items to the relevant sections (table 6).

The items of ePAQ-VAS were arranged into four sections: generic, AAA, CAD and lower limb (LL) vascular conditions. A single LL section was developed as common themes were identified for conditions affecting the legs, regardless of whether the underlying pathology was venous or arterial. An inclusive approach to development was used and a comprehensive questionnaire was produced with 168 questions (see figure 2 for an overview of the process to develop ePAQ-VAS).

ePAQ-VAS was presented to 19 vascular patients. Overall, the response was positive; the participants felt the generic, and the relevant disease specific were comprehensive, fit for purpose and potentially useful. There was little consistency in items that participants found difficult and no individual item was identified with which most participants had difficulty.

Discussion included the use of abbreviations, font size and contrast between text and background, response options and scales, electronic format versus paper format, relevance to patients and clinicians, the use of free-text boxes and the language and wording used, when and how to use the skip button, repetition of items and subject matter and the possibility of emotional distress associated with questions about the possibility of deterioration or death.

Based on the findings from the face validity exercise, and input from the vascular PROMs group, further

revisions were made in an iterative process, culminating in the development of ePAQ-VAS. The structure of the questionnaire is illustrated in figure 3. Fifty-nine items were eliminated for overlap; these include questions asking about common symptoms experienced across most vascular conditions. Five items were added based on suggestion from clinicians. Generic items for all respondents were presented in the first section and include questions about pain, altered sensation, weakness, weight/height, smoking habit, previous medical history and regular medication. This information was deemed important for assessment of vascular conditions both by patients and clinicians.

The next three sections are condition specific relating to CAD, AAA and LL vascular disease sections. These sections are further divided into scales. There are 55 items within eight scales and the remainder of questions do not contribute to scales but are kept due to their clinical relevance. The eight scales are part of the condition-specific sections and include CAD-related anxiety, impact of CAD on activities of daily living (ADLs), AAA-related anxiety, impact of AAA on activities of ADL, PAD symptoms, VLU symptoms, VV symptoms and impact of LL vascular disease on ADL. Individual items, scales and sections of ePAQ-VAS in its initial version can be viewed on https://demo-questionnaire.epaq.co.uk/home/project?id=VASC_1.6&page=1.

The evidence used to develop each item in ePAQ-VAS is made explicit in table 5; this table show whether the source for the item is the qualitative study, reviews or consensus study.

## DISCUSSION

This study documents stages undertaken to develop ePAQ-VAS and the conceptual framework underpinning this new tool for use in undifferentiated vascular populations. The main strength of this new instrument is that it can be used as a holistic clinical assessment tool that can be completed by patients before meeting the vascular surgeon in the clinic. The information generated can be used to help shared decision making by focusing on patient priorities. This tool has the further advantage of being an electronic online PROM since it can be used to monitor impact of the disease and/or interventions overtime. Furthermore, this instrument is preference based, unlike the identified vascular PROMs[4–8]; once further validated, the disease-specific scales can be used to generate utility values either by mapping to the values of a generic utility measure or by further utility studies.[18]

This instrument has been developed in line with FDA guidelines for developing PROMs.[3] Items were developed based on themes extracted from primary qualitative data, systematic qualitative evidence synthesis and clinicians' consensus exercise.[9–13] We have made efforts through purposive sampling to ensure that we have included diverse demographic groupings in the primary research, and this is augmented by the inclusion of systematic reviews that include evidence gathered in national and international studies. Another strength of this study is that the qualitative evidence in the review and the primary study included patients at different stages of their disease. The data collected included the impact of disease, including symptoms, on daily living and the impact of diagnosis and treatment on the daily living. The vascular clinicians' input into developing and rating the items was sought, and new items were incorporated based on recommendation from 25 vascular clinicians.

The work of developing individual items and their assignment to putative scales and sections was based both on the framework of existing PROMs[4–8] and on input from vascular clinicians. In this stage of the ePAQ-VAS development, an inclusive approach was chosen, and all relevant items were incorporated except for those with clear repetition. The main limitation of this draft version of ePAQ-VAS is that it is long and potentially repetitive; it is expected that factor analysis and psychometric testing will lead to a reduction in the number of individual items and will confirm (or refute) the putative scales identified in the current version. Furthermore, skipping rules embedded within the questionnaire will only present the items relevant to the patient completing the online instrument.

Another limitation is that ePAQ-VAS only cover the five main vascular conditions, and it might not be relevant to patients with other vascular disease. However, including all vascular conditions in one instrument is not possible, and the evidence to include only these conditions was based on input from clinicians treating vascular disease. As stated by the FDA,[3] a fundamental consideration in the development of PROMs is the adequacy of item generation. Due to the heterogeneous nature of vascular disease, it was not straightforward to identify what exactly should be measured when developing and defining the initial conceptual framework for the ePAQ-VAS. To this end, as recommended by the FDA, the initial conceptual framework was based on information gathered from reviews of the literature, patients and expert opinion.

The findings of the qualitative study indicated an overlap in patient experiences of the various conditions. However, there was also a clear difference in how each condition impacted on different aspects of quality of life. There were conditions with many physical symptoms and others with none. This demonstrated that while it may be possible to develop a PROM for use across a variety of vascular conditions, it would also need to include condition-specific items to fully capture the impact and clinically relevant information for each disease or condition. A further limitation is that the face validity study was not able to examine the comprehensiveness of ePAQ-VAS since it covered multiple conditions, and it was difficult to expect from any of the patient groups interviewed to comment on diseases they have not experienced. Therefore, they only commented on the generic questions and the disease-specific items relating to their condition.

In conclusion, ePAQ-VAS is a multidimensional measure developed for use in a range of vascular conditions. It is a single electronic tool, covering most vascular conditions. This is important for those patients presenting with mixed symptoms or multiple conditions. The items in ePAQ-VAS can capture information about disease symptoms, HRQoL, comorbidities, medical history and other relevant healthcare issues. This type of information can aid communication between healthcare professionals and patients and support shared decision making. The electronic format may make it easier to monitor patients over time, especially those with chronic conditions and those treated with lifestyle modification or conservatively. Based on methods used in its construction, this tool has a strong degree of content validity; however, further psychometric testing for reliability, responsiveness and validity is needed. Once this electronic PROMs is validated, it can be used as an outcome measure in clinical practice and research.

**Acknowledgements** The authors would like to thank the UK Vascular PROMs Group and Cardiovascular Research Patient Panel at Sheffield Teaching Hospitals National Health Service (NHS) Foundation Trust.

**Collaborators** UK Vascular PROMs Group: Alison Abbott; Ayman Badawy; Nasim Akhtar; Faisal, Alam; Wissam Al-Jundi; Matthew Armon; Lukla Biasi; Matt Bown; Bruce Campbell; Tommaso Donati; Tony Fox; Andrew Gordon; Linda Hands; Gary Hicken; Peter Holt; Shelley Jackson; Brenda King; Talia Lea; Greg McMahon; John Mosley; Sanjay Patel; Gaynor Radley; Harjeet Rayt; Rob Sayers; and Paul Tisi.

**Contributors** JM, GJ, AA and SR designed the study. AA, PP, EL, SR and GJ performed the analysis of the data and theme generation. ST and SN helped with consensus study and qualitative study recruitment. AA wrote the paper, and all authors reviewed and edited the paper. All authors approved the manuscript.

**Funding** This manuscript presents independent research funded by the National Institute for Health Research under the Programme Grants for Applied Research (RP-PG-1210–12009).

**Competing interests** SR is a shareholder in ePAQ Systems Ltd, an NHS spin-out technology company, the majority shareholder being Sheffield Teaching Hospitals.

**Patient and public involvement** Patients and/or the public were involved in the design, or conduct, or reporting, or dissemination plans of this research. Refer to the Methods section for further details.

**Patient consent for publication** Not required.

**Ethics approval** Ethical approval was obtained from Yorkshire & Humber NRES committee – Bradford Leeds (REC Number: 14/YH/1117) on 25 September 2014.

**Provenance and peer review** Not commissioned; externally peer reviewed.

**Data availability statement** No data are available.

**ORCID iDs**
Ahmed Aber http://orcid.org/0000-0002-5707-2430
Jonathan Michaels http://orcid.org/0000-0002-3422-7102

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
