## [Reviewer comments · BMJ Open]

ARTICLE DETAILS

TITLE (PROVISIONAL)	Mixed methods study to develop the content validity and the conceptual framework of the electronic patient-reported outcome measure for vascular conditions
AUTHORS	Aber, Ahmed; Phillips, Patrick; Lumley, Elizabeth; Radley, Stephen; Thomas, Steve; Nawaz, Shah; Jones, Georgina; Michaels, Jonathan

VERSION 1 - REVIEW

REVIEWER	Ying Wei Lum Johns Hopkins Hospital
REVIEW RETURNED	21-Oct-2019

GENERAL COMMENTS	Congratulations on a very well written and clearly delineated paper. I think my only suggestion is that you explain how this electronic questionnaire will be administered and how it will be incorporated into the records, particularly if patients seek treatment at different NHS trusts (or is that not allowed?).
---

REVIEWER	Derek Kyte University of Birmingham, UK
REVIEW RETURNED	25-Feb-2020

GENERAL COMMENTS	This is an important manuscript addressing a clear need for a consolidated vascular ePROM. It is good to see such attention/importance placed on addressing content validity and the development of a conceptual framework. As outlined below, my main issues relate to the lack of detail around the methodology of the component studies, which makes it somewhat difficult to fully assess the quality of the work and does rather undermine the findings/recommendations presented in the discussion/conclusion. It is clear that there are a number of published reports which describe the individual studies in detail – these are referenced, albeit a little sporadically. However, I feel more detail is needed within this manuscript to more successfully pull the separate studies together into a coherent package that readers can efficiently scrutinise. Particular comments: • Important to signpost in the title that content validity was evaluated.
---

	 • It is convention to use 'PROMs' where plural acronym is required. • Pg. 7 - There were few details around the search strategy used to identify and appraise existing PROMs, therefore it is not possible to ascertain if this search was appropriate/comprehensive. Similarly, there are no details outlining the steps taken during the quality assessment, therefore it is not possible to determine if the methods adhered to the referenced standards (FDA and COSMIN methodology). These sections should include the references to the published reports where appropriate. It would also be useful to include granular details around search strategy/results and appraisal in a supplementary appendix. • It was not clear if trained/skilled group interviewers were used in the qualitative aspects, nor if topic guides were used? Also, aspects surrounding coding (e.g. independent coding, member validation) or saturation were not consistently described. • Slightly more detail is needed to fully describe each of the six overarching themes presented on pg.12-13 (symptoms, impact on physical function, social impact, psychological impact, financial impact and lifestyle). • It was surprising that no patients were involved in the consensus exercise, this may need addressing in the limitations. • Owing to a marked lack of detail, it was not possible to determine if the 19 patients involved in the cognitive interviews (face validity) formed a sample representing the target population. It was not clear if all items were viewed in their final form. It was not possible to tell if comprehensibility of the PROM instructions, items, response options, and recall period were investigated. It was assumable that the comprehensiveness of the PROM was explored but this was not clearly described. • The results section appears to be missing detail around the characteristics of the participants in each phase. • It is assumable that there were no problems identified during the cognitive interviews with regards comprehensibility, comprehensiveness or relevance, or that problems were appropriately addressed, but this was not clearly described. • The discussion section appears to missing a summary of study limitations. Thank you for the opportunity to review this well written manuscript.
--	--

REVIEWER	Xinhua Yu University of Memphis US
REVIEW RETURNED	25-Feb-2020

GENERAL COMMENTS	This study presents the stages of developing a patient report outcome questionnaire. The overall writing is clear and the paper is well organized. There are a few minor issues that may enhance the readability of this paper.  1) For some disease conditions, such as AAA, CAD, there is no physical symptoms to measure, which means patients will have little sense about what is going on in their body. Then there are some concerns that this questionnaire can yield some useful information specific on these conditions. 2) page 15, the author mentioned triangulation approach, and gave an example in the appendix. However, it would be more informative to readers if there is more details in terms how to conduct it and how
---

	the results from triangulation approach used in determining questionnaire items. 3) This questionnaire is fairly long, and it would be helpful to present some quantitative information about how much time is needed for a patient to fill in the questionnaire, and assess the preliminary item response patterns within the questionnaire among 19 test patients. 4) the questionnaire is a mixture of quantitative items (likert scale or binary) and qualitative (open) items. How are the results presented and how can users analyze them? How does a summary report look like? 5) what is the next step? I think it needs a pilot test on a larger sample base to evaluate the questionnaire quantitatively.
--	--

VERSION 1 – AUTHOR RESPONSE

Source	Comments	Responses
Reviewer 1	Congratulations on a very well written and clearly delineated paper. I think my only suggestion is that you explain how this electronic questionnaire will be administrated and how it will be incorporated into the records, particularly if patients seek treatment at different NHS trusts (or is that not allowed?).	Thank you for this important point. ePAQ can be accessed online by patients. Each patient receives a unique voucher code that they can use it to access and complete the questionnaire at home or in the outpatient clinic. The server of ePAQ is hosted and integrated with NHS N3-based informatics systems. Other ePAQ questionnaires such as ePAQ-Pelvic floor and ePAQ- pre-assessment are in clinical use in different NHS hospitals. The data collected from these PROMs are hosted on a server housed by Sheffield Teaching Hospitals. ePAQ Ltd. is an NHS spin-out technology company and the patient data collected by the company can be linked to the unique NHS number of each patient; and although there is a lack of integrated digital infrastructure in the NHS; the technology is available for future use to link records collected by different NHS providers.
Referee: 2	My main issues relate to the lack of detail around the methodology of the component studies, which makes it somewhat difficult to fully assess the quality of the work and does rather undermine the findings/recommendations presented in the discussion/conclusion. It is clear that there are a number of published reports which describe the individual studies in detail – these are referenced, albeit a little sporadically. However, I feel more detail in needed within this manuscript to more	Thanks for the comment. The paper has been updated to add extra information about the systematic reviews, the primary qualitative study, the consensus study and the face validity study.

Source	Comments	Responses
	successfully pull the separate studies together into a coherent package that readers can efficiently scrutinise.	
	Important to signpost in the title that content validity was evaluated.	The title has been updated to include this.
	It is convention to use 'PROMs' where plural acronym is required.	PROMs have been used throughout
	Pg. 7 - There were few details around the search strategy used to identify and appraise existing PROMs, therefore it is not possible to ascertain if this search was appropriate/comprehensive . Similarly, there are no details outlining the steps taken during the quality assessment, therefore it is not possible to determine if the methods adhered to the referenced standards (FDA and COSMIN methodology). These sections should include the references to the published reports where appropriate. It would also be useful to include granular details around search strategy/results and appraisal in a supplementary appendix.	Many thanks for this comment. Further detail is now provided about the search strategies of the systematic review and the psychometric appraisal criteria. For PAD review COSMIN checklist was used, however for AAA, CAD, VVs, VLU appraisal criteria were used based on FDA guidelines, University of Oxford PROMs guidance and COSMIN. Similar criteria have been previously utilised by other groups.
	It was not clear if trained/skilled group interviewers were used in the qualitative aspects, nor if topic guides were used? Also, aspects surrounding coding (e.g. independent coding, member validation) or saturation were not consistently described.	Thank you. The method section has been updated to add the extra information requested. The interviews were conducted by trained qualitative researchers, coding was done in accordance to framework analysis and themes were identified until saturation was achieved. All themes were checked by an independent researcher and any differences were adjusted by involving a senior author.

Source	Comments	Responses
	Slightly more detail is needed to fully describe each of the six overarching themes presented on pg.12-13 (symptoms, impact on physical function, social impact, psychological impact, financial impact and lifestyle).	Thank you. This section has been expanded to include extra information in addition to the table provided.
	It was surprising that no patients were involved in the consensus exercise, this may need addressing in the limitations.	The consensus exercise was to examine the clinical relevance of the data and to add items that the clinicians felt were important but not included in the primary version. No item was dropped based on this study and only extra items were added. This was to ensure that ePAQ-VAS is clinically relevant and it useful to the clinicians so that they will use it regularly in their clinics.
	Owing to a marked lack of detail, it was not possible to determine if the 19 patients involved in the cognitive interviews (face validity) formed a sample representing the target population. It was not clear if all items were viewed in their final form. It was not possible to tell if comprehensibility of the PROM instructions, items, response options, and recall period were investigated. It was assumable that the comprehensiveness of the PROM was explored but this was not clearly described.	The method section describing this study have been updated to provide the extra information described in this comment. Also, the discussion section discussed issues around the limitations of this face validity study.
	The results section appears to be missing detail around the characteristics of the participants in each phase.	Many thanks for this comment. This information has been provided in the primary study and as per your comment we have added the participant information in the revised version of this manuscript.
	It is assumable that there were no problems identified during the cognitive interviews with regards comprehensibility, comprehensiveness or	The results and discussion sections have been updated to address this comment. Since ePAQ-VAS covers five different vascular disease categories, the participants could only comment on the comprehensiveness of the generic and the relevant disease specific category. The response was positive; however, the limitation remains that there was

Source	Comments	Responses
	relevance, or that problems were appropriately addressed, but this was not clearly described.	no single group that could comment on the entire instrument. This point was mentioned in the limitations of the study.
	The discussion section appears to missing a summary of study limitations.	Thank you. This section has been updated and the limitation subsection is signposted and expanded.
Referee 3	For some disease conditions, such as AAA, CAD, there is no physical symptoms to measure, which means patients will have little sense about what is going on in their body. Then there are some concerns that this questionnaire can yield some useful information specific on these conditions.	Thank you. The result section of the manuscript has been updated to include this information.
	page 15, the author mentioned triangulation approach, and gave an example in the appendix. However, it would be more informative to readers if there is more details in terms how to conduct it and how the results from triangulation approach used in determining questionnaire items.	The result section has been updated to give more information about the use of triangulation. This exercise was done across AAA, PAD, CAD, VLU, VVs. The aim was to identify all relevant themes and avoid repetition when generating items and question. No themes were disregarded based on this.
	This questionnaire is fairly long, and it would be helpful to present some quantitative information about how much time is needed for a patient to fill in the questionnaire, and assess the preliminary item response patterns within the questionnaire among 19 test patients.	Thank you for this comment. There are skipping rules embedded in this electronic questionnaire. These ensure that only relevant questions are presented to the patients. The questionnaire has been tested in a large survey, the results of this survey have been presented in another paper that is in press currently. The data from this study reported that the mean time to complete ePAQ-VAS in the clinic was 12:51 minutes (Median, 09:14 minutes) and online prior to the clinic appointment was 36:51 minutes (Median 30:44 minutes), the difference in completion time is likely due to availability of help from researchers in clinics to complete ePAQ-VAS.
	the questionnaire is a mixture of quantitative	A demonstration version of the final version of ePAQ can be viewed at: http://demo-

Source	Comments	Responses
	items (likert scale or binary) and qualitative (open) items. How are the results presented and how can users analyze them? How does a summary report look like?	questionnaire.epaq.co.uk/home/project?id=aaa_1.0&page=1 A copy of final report is added to the supplementary data.
	What is the next step? I think it needs a pilot test on a larger sample base to evaluate the questionnaire quantitatively	Thank you for your comment. This study has been completed and its results are reported in a paper accepted by BJS and currently in press.

VERSION 2 – REVIEW

REVIEWER	Xinhua Yu University of Memphis, US
REVIEW RETURNED	19-Mar-2020

GENERAL COMMENTS	No additional comments. The authors addressed reviewers' concerns.
--